# Antimicrobial Efficacy of Quercetin against *Vibrio parahaemolyticus* Biofilm on Food Surfaces and Downregulation of Virulence Genes

**DOI:** 10.3390/polym14183847

**Published:** 2022-09-14

**Authors:** Pantu Kumar Roy, Sung-Hee Park, Min Gyu Song, Shin Young Park

**Affiliations:** 1Institute of Marine Industry, Department of Seafood Science and Technology, Gyeongsang National University, Tongyeong 53064, Korea; 2World Institute of Khimchi, Gwangju 61755, Korea

**Keywords:** *Vibrio parahaemolyticus*, quercetin, biofilm, shrimp, crab, gene expression

## Abstract

For the seafood industry, *Vibrio parahaemolyticus*, one of the most prevalent food-borne pathogenic bacteria that forms biofilms, is a constant cause of concern. There are numerous techniques used throughout the food supply chain to manage biofilms, but none are entirely effective. Through assessing its antioxidant and antibacterial properties, quercetin will be evaluated for its ability to prevent the growth of *V. parahaemolyticus* biofilm on shrimp and crab shell surfaces. With a minimum inhibitory concentration (MIC) of 220 µg/mL, the tested quercetin exhibited the lowest bactericidal action without visible growth of bacteria. In contrast, during various experiments in this work, the inhibitory efficacy of quercetin without (control) and with sub-MICs levels (1/2, 1/4, and 1/8 MIC) against *V. parahaemolyticus* was examined. With increasing quercetin concentration, swarming and swimming motility, biofilm formation, and expression levels of related genes linked to flagella motility (*flaA* and *flgL*), biofilm formation (*vp0952* and *vp0962*), and quorum-sensing (*luxS* and *aphA*) were all dramatically reduced (*p* < 0.05). Quercetin (0–110 μg/mL) was investigated on shrimp and crab shell surfaces, the inhibitory effects were 0.68–3.70 and 0.74–3.09 log CFU/cm^2^, respectively (*p* < 0.05). The findings were verified using field emission scanning electron microscopy (FE-SEM), which revealed quercetin prevented the development of biofilms by severing cell-to-cell contacts and induced cell lysis, which resulted in the loss of normal cell shape. Furthermore, there was a substantial difference in motility between the treatment and control groups (swimming and swarming). According to our findings, plant-derived quercetin should be used as an antimicrobial agent in the food industry to inhibit the establishment of *V. parahaemolyticus* biofilms. These findings suggest that bacterial targets are of interest for biofilm reduction with alternative natural food agents in the seafood sector along the entire food production chain.

## 1. Introduction

Nutritious and tasty aquatic products are susceptible to oxidation and bacterial contamination in planned transportation. This situation damages the taste of aquatic products and seriously threatens food safety. Since seafood is freshly used, the food quality is very high, but, in time, its quality will decline and become unfit for consumption. The quality and safety of seafood is a major challenge facing its own seafood-related industry in food sciences, and mostly in fisheries and aquaculture research departments. Fish is an essential nutrient in the human diet and is also present in the global aquatic product industry for consumers. Such products are prone to oxidation and bacterial contamination in organized transportation, which not only destroys the taste of products, but also poses a very serious threat to food safety [1]. Seafood is frequently contaminated with the Gram-negative bacterium *Vibrio parahaemolyticus* [2]. During infection, it forms a biofilm, which is a layer of self-produced proteins, lipids, and polysaccharides that covers the surface of the host [3]. A crucial aspect of the pathogenesis is the production of biofilm, which might increase resistance to harmful circumstances and medications. According to studies by Han et al. [4] and Almohamad et al. [5], over 60% of outbreaks by *V. parahaemolyticus* biofilm occurred by consuming contaminated seafoods. Infections with *V. parahaemolyticus* typically have self-limiting symptoms (e.g., vomiting, diarrhea, fever, nausea, chills, headaches, and watery stools [6,7]). Although uncommon, this bacterium can cause septicemia, necrotizing fasciitis, wound infections, and even death [8,9]. Because of this, *V. parahaemolyticus* contamination poses a threat to the aquaculture industry, the food industry, and public health.

The World Health Organization reported that O3:K6 serotypes and their variants are the most common strains associated with foodborne diseases, with *V. parahaemolyticus* being the most often encountered bacterial gastroenteritis, associated with the consumption of seafood products globally [10]. One of the main issues for food safety and public health has been the prevalence of *V. parahaemolyticus* in the world. According to the CDC [11], *V. parahaemolyticus* causes 45,000 illnesses annually in the USA and is the most often reported in *vibrio* infections (https://www.cdc.gov/vibrio/faq.html, accessed on 15 June 2022) [7]. Currently, standard methods for preventing and treating *V. parahaemolyticus* contamination and infection, such as antibiotics and chemical disinfectants, are crucial [12,13]. However, studies indicate that *V. parahaemolyticus* clinical isolates and environmental isolates both show rising antibiotic resistance globally [14,15,16]. Because of the limitations of current control systems, other techniques of preventing bacterial contamination and infections are constantly being studied [17,18].

In comparison to their planktonic relatives, biofilms are a million times more resistant to all antimicrobial treatments [1,19]. Because of this, removing biofilm with common medicines and cleaning supplies may be challenging [4]. Aggressive chemicals, such as sodium hydroxide or sodium hypochlorite, are frequently employed in the food sector to reduce the negative impacts of biofilm [20]. However, such methods might damage the environment by corroding equipment and materials [21,22]. Therefore, it is essential to create a successful strategy that can manage and eradicate bacterial biofilm. The term “biofilm” refers to bacterial growth that defends itself by routinely embedding cells in extracellular polymeric substances (EPS) as opposed to free-living bacterial cells [7,23]. This increases the bacteria’s ability to survive acquaintance to antimicrobial agents [7,24]. A number of biofilm-related genes regulate the continuous, dynamic processes that lead to the formation of biofilms, including cell attachment, EPS synthesis, resource capture, detachment, and dispersal. According to studies [14,25], *V. parahaemolyticus* can form biofilms on a variety of biotic or abiotic surfaces and interfaces, including seawater and marine organisms (shrimp, fish, crab, shellfish, etc.) [7]. This contamination of the sea and seafood leads to cross-contamination during the processing or preparation of food [7,26]. Cross-contamination may be a significant source of human diseases, according to reports [27]. The development of biofilms on or in seafood may play a significant role in the spread of *V. parahaemolyticus* and the subsequent illnesses [28]. In order to reduce contamination and infections caused by *V. parahaemolyticus*, biofilm serves as a significant target.

Bacteria are protected from physical harm, desiccation, and antibiotics by microbial biofilms [29]. Previous studies reported food-borne pathogens persist as biofilms on foods (shrimp, crab shell, and lettuce) and food contact surfaces (e.g., plastic, steel, glass, and rubber) and have an impact on the quantity, quality, and safety of food products [30,31,32,33,34]. Additionally, they destroy surfaces and equipment, contaminate food on a constant basis, pose a significant risk to public health, and their control is a significant barrier in the food production chain [35]. To prevent foodborne infections, natural plant extracts and antimicrobial compounds are typically regarded as secure, efficient, and environmentally friendly [36,37]. With a broad range of activity against numerous bacterial and fungal infections, plant extracts have long been used extensively for food safeguarding and disease anticipation [7,37,38].

The use of inhibitory compounds that interfere with quorum sensing (QS) is one of the preventative techniques for improving food quality and safety [39,40]. QS in a number of bacteria can be disrupted by phenolic chemicals generated from plants [39]. Plant compounds are an alternative control method against *V. parahaemolyticus* biofilms, and one of the most investigated flavonoid molecules having functional characteristics in this context is quercetin. Flavonoids have become well-known for having anti-inflammatory, antioxidant, antibacterial, and anticancer properties [41] in addition to their potential QS system inhibitory properties [42,43]. Many fruits and vegetables, including apples, tea, onions, red grapes, berries, tomatoes, and tea, contain quercetin, a flavonoid-based compound [44]. Due to its anti-inflammatory, anticancer, and neuroprotective properties, it has a wide range of applications [45,46]. The ability of flavones and flavonoids to scavenge free radicals inside cells make them recognized as having strong antioxidant activity. As a subclass of polyphenolic substances with a wide range of chemical characteristics, flavonoids are a ubiquitous component of plants. Of all the flavonoids, quercetin is one of the most effective antioxidants [47,48]. More than 4000 different flavonoids, including flavonols, flavones, flavanones, catechins, anthocyanidins, isoflavones, dihydroflavonols, and chalcones, have been classified as part of the major flavonoid group [47]. The food business uses organic antioxidants to preserve the product’s color and nutritional content. Studies on the application of flavonoids in various industrial sectors have grown in number recently [47]. Similar to how they may be used in food, textiles, leather, metallurgy, medicine, and agriculture, these substances may also be used for their antioxidant capabilities [47]. Quercetin is, therefore, a common source for the food and pharmaceutical industries [47]. Therefore, there is an increasing demand for biodegradable polymers (fibers) that are effective against microorganisms. Quercetin-(Q)-loaded polylactide-based polymers (fibers) were used as antibacterial effects against *Staphylococcus aureus*, *Escherichia coli*, and *Klebsiella pneumoniae* [49]. The nanoparticle form of quercetin (nanoquercetin) is composed of a polyaspartic acid-based polymer micelle encapsulated with quercetin; colloidal in nature, they were used for inhibiting intracellular polyglutamine aggregation in cellular and animal models of Alzheimer’s diseases [50]. Polymer micelle-encapsulated quercetin also has anticancer properties, and is used in the development of biodegradable nanoparticles, and enhanced delivery of quercetin by encapsulation in a surfactant polymer as an antisolvent process [51,52]. Owing to its three-ring structure with five hydroxyl groups, it possesses especially strong antioxidant capabilities [33,41,45]. Antioxidants can reduce oxidative stress and prevent biofilm formation by scavenging reactive oxygen species (ROS) accumulated in bacterial cells [33,41]. As a result, antioxidants are potent antibiofilm agents [45,53] as it is one of the primary processes by which oxidative stress induces bacteria to develop biofilm as a survival strategy. Additionally, it has already been demonstrated that quercetin has antibacterial properties against both Gram-positive and Gram-negative bacteria [44], including *S. aureus* [44,54], *E. coli* [44,55], and *Pseudomonas aeruginosa* [44,56].

The antimicrobial action of quercetin against *V. parahaemolyticus*, however, has not been specifically investigated in any investigations. In the present work, the ability of quercetin to suppress *V. parahaemolyticus* biofilm formation, including flagella motion, and its effects on virulence and QS gene expression were assessed.

## 2. Materials and Methods

### 2.1. Bacterial Strain Culture and Growth Conditions

*Vibrio parahaemolyticus* strain from the American Type Culture Collection (Manassas, VA, USA) (ATCC27969) was collected and used for the biofilm-forming assays. The bacteria were cultured in tryptic soy broth (TSB, BD Difco, Franklin Lakes, NJ, USA) with 2.5% NaCl at 30 °C for 24 h followed by another sub-culture at 18 h [57]. The culture was centrifuged (11,000× *g* for 10 min) and washed two times with phosphate buffered saline (PBS; Oxoid, Basingstoke, England). After that, peptone water (PW; Oxoid, Basingstoke, England) was added to the final bacterial solution to dilute it until it contained 10^5^ log CFU/mL of bacteria. The formation of biofilms on surfaces of crab and shrimp was then accomplished using these inoculums (10^5^ CFU/mL).

### 2.2. Preparation of Samples (Crabs and Shrimp)

With few modifications, sample preparation was done as explained in our earlier investigations [57]. The shrimp (*Penaeus monodon*) and crabs (*Corystes cassivelaunus*) were bought at a nearby grocery store at Tongyeong local market. Using a sterile scalpel, crab and shrimp shells were sliced into 2 × 2 cm^2^ coupons. Following the removal of any leftover meat, the shells were cleaned with sterile distilled water (DW). Before being inoculated with *V. parahaemolyticus*, the coupons were sterilized by being exposed to UV-C light for 15 min on each side to remove background microflora [57]. The coupons were dipped into 10 mL of TSB, inoculated with bacteria (10^5^ log CFU/mL), and then incubated for 24 h at 30 °C without shaking to test for the further experiment.

### 2.3. Quercetin Preparation and Determination of Minimum Inhibitory Concentration (MIC)

For our study quercetin (Q-4951) was collected from Sigma-Aldrich (St. Louis, MO, USA). After being dissolved in dimethyl sulfoxide (DMSO, Sigma-Aldrich, St. Louis, MO, USA), the product was used to create a stock solution with a concentration of 1 mg/mL. The MIC was verified and very slightly modified from the previous study [33]. A two-fold serial dilution approach using TSB was used to establish the minimum inhibitory concentration (MIC) of quercetin against *V. parahaemolyticus*. A total of 100 µL of quercetin serially diluted with TSB and 100 µL of bacterial suspension (10^5^ log CFU/mL) were combined in 96-well plates (Corning Incorporated, Corning, Inc., Corning, NY, USA). Each well had a total amount of 200 µL. A microplate reader (Spectra Max 190, Sunnyvale, CA, USA) was used to measure absorbance (600 nm) while the plates were kept in a 30 °C incubator for 24 h. After an overnight incubation at 30 °C, aliquots (100 µL) taken from the wells that had no discernible growth were plated on Vibrio CHROMagar plates and the number of colonies counted. Triplicates of this experiment were run. The MIC of quercetin was 220 µg/mL. For further experiments in this study, sub-MICs of quercetin were used.

### 2.4. Analysis of Motility

Motility experiments were carried out in this study with minor modifications from those previously published [33]. This test was conducted to verify the effect of quercetin on the two forms of *V. parahaemolyticus* motility (swimming and swarming). Bacto agar (BD Dicfo, Franklin Lakes, NJ, USA) was mixed with TSB at a rate of 0.3% and 0.5% to provide the media for the swimming and swarming studies, respectively, and incubated at 30 °C for 13 h for swimming and 48 h for swarming. Each plate was filled with the autoclaved medium. Quercetin was added after the autoclaved medium and thoroughly mixed in before it solidified. The diameter of bacterial movement through agar was measured in millimeters (mm).

### 2.5. Biofilm Formation and Detachment Process

With slight adjustments, the procedure was carried out as previously described [33]. The MIC in this study was 220 µg/mL, and the inhibiting effect of biofilm was seen at sub-MIC levels, which may not have killed the bacteria, but affected their virulence factor. Control (without quercetin), 1/8, 1/4, and 1/2 MIC concentrations were used in this study. In a 50 mL conical tube 10 mL TSB with food surfaces, quercetin was added (0, 1/8, 1/4, and 1/2 MICs), and 100 µL of bacterial suspension (10^5^ log CFU/mL). They were then thoroughly combined with a vortex mixer (Scientific Industries, SI-0256, Bohemia, NY, USA) and incubated for 24 h at 30 °C to form biofilms. To get rid of bacteria that had somewhat attached to the surfaces after the biofilm formation, the coupons were washed twice with distilled water (DW) [33,41]. Each washed coupon (shrimp and crab) was placed in a sterile stomacher bag with 10 mL of peptone water (PW; BD Diagnostics, Franklin Lakes, NJ, USA), and processed using a Stomacher (Bag Mixer; Interscience, Saint Nom, France) at the highest speed of 4 for two min to release the bacteria that form biofilms on the coupons [34]. This bacterial sample was serially diluted before being placed in Vibrio CHROMagar plates as an inoculum. The number of colonies on the plates was counted after they had been kept in a 30 °C incubator for 24 h. After subtracting the populations of each concentration (0, 1/8, 1/4, and 1/2 MIC) from the populations of each group, we were able to calculate the inhibition values at log CFU/cm^2^.

### 2.6. Confirmation of Biofilms Inhibition by FE-SEM

A *V. parahaemolyticus* strain known for its high biofilm-forming ability was used to test the biofilm-forming ability using FE-SEM. According to our earlier published work, the samples were prepared [57]. Briefly, the samples were washed in PBS and placed in 6-well dishes, fixed with 2.5% glutaraldehyde, and stored at room temperature for 4 h, and after that treated with ethanol (50, 60, 70, 80, 90% for 15 min serially) and 100% for 15 min two times. Then, the treated samples were dehydrated with soaking (33, 50, 66, and 100% hexamethyldisilazane in ethanol) for 15 min serially. The samples were platinum sputed-coated and observed by FE-SEM (Hitachi/Baltec, S-4700, Tokyo, Japan) [32].

### 2.7. Relative Gene Expression by Real-Time PCR (RT-PCR)

With a few minor adjustments, the experiment was carried out as previously described [33]. The test was carried out to confirm quercetin’s impact on *V. parahaemolyticus* pathogenicity and quorum-sensing gene expression. After biofilm formation on coupons, biofilm cells were collected for RNA extraction. Total RNA was collected using the RNeasy Mini kit (Qiagen, Hilden, German) followed by the manufacturer’s protocol. Using a Maxime RT PreMix (Random Primer) kit (iNtRON Biotechnology Co., Ltd., Seoul, Gyeonggi-do, Korea), cDNA was produced after the RNA yield and purity were assessed using a spectrophotometer at 260/280 nm and 260/230 nm (NanoDrop, Bio-Tek Instruments, Chicago, IL, USA) [58]. Table 1 lists the primers. The housekeeping gene was 16S rRNA. In a total volume of 20 µL, the cDNA sample was combined with the appropriate primers and Power SYBR Green PCR Master Mix (Applied Biosystems, Thermo Fisher Scientific, Warrington, UK). A CFX Real-Time PCR System (Bio-Rad, Hercules, CA, USA) was used to perform the RT-PCR analysis. Utilizing 2X Real-Time PCR Master Mix and 1 µL of cDNA as a template, RT-qPCR was carried out. A CFX Real-Time PCR System was used to conduct the real-time PCR. Initial denaturation for the PCR reaction took place at 95, 50, and 72 °C for 20 s each [59,60,61]. After PCR cycling was complete, we collected cycle threshold (Ct) values to confirm the specificity and conducted 2^−^^△△Ct^ method analysis [62,63,64].

### 2.8. Statistical Analysis

The experiments were performed at least three times. All data were expressed as mean ± standard error of mean (SEM). Statistical significance was set at *p* < 0.05 when Duncan’s multiple-range test and an ANOVA were performed using SAS software version 9.2 (SAS Institute Inc., Cary, NC, USA) to determine the significance.

## 3. Results

### 3.1. Swimming and Swarming Motility Assays

For the formation of biofilms, bacterial flagella must be mobile. *V. parahaemolyticus* flagella can be verified by swimming and swarming assays, in particular. The impact of quercetin on inhibiting *V. parahaemolyticus* motility is depicted in Figure 1 and Figure 2. Quercetin reduced *V. parahaemolyticus* motility by 18 and 79%, respectively, in the swimming experiment when compared to the control at 1/8 and 1/2 MIC. Figure 2 depicts the quercetin’s inhibition of *V. parahaemolyticus*. Quercetin thereby reduced *V. parahaemolyticus* motility by 14 and 57% at 1/8 and 1/2 MIC, respectively. Thus, in this experiment, as quercetin concentration increased, swimming and swarming motility became more inhibited. Particularly in comparison to the control group, motility was significantly different with 1/2 MIC of quercetin.

### 3.2. Eradication Effect of Food Additive Quercetin on Shrimp and Crab Shell Surfaces against V. parahaemolyticus

The *V. parahaemolyticus* biofilm on shrimp coupons is shown, in Figure 3, to be inhibited by quercetin. As quercetin content increased, the biofilm-inhibiting impact also grew. The *V. parahaemolyticus* biofilm inhibition values on the shrimp surface were 0.68, 1.43, and 3.70 log CFU/cm^2^, respectively, at quercetin quantities of 1/8, 1/4, and 1/2 MIC. Comparing these values to the control and other MIC groups, they were significantly (*p* < 0.05) suppressed at 1/2 MIC. On crabs, *V. parahaemolyticus* biofilm is shown, in Figure 4, to be inhibited by quercetin. The *V. parahaemolyticus* biofilm inhibitory values were 0.74, 1.40, and 3.09 log CFU/cm2 at 1/8, 1/4, and 1/2 MIC quercetin concentrations, respectively. Compared to the control and other MIC groups, 1/2 MIC significantly inhibited biofilm formation (*p* < 0.05).

### 3.3. Visual Confirmation of Biofilm Reduction by Quercetin under FE-SEM

The visual confirmation of biofilm inhibition by quercetin is shown in Figure 5. Biofilms were architecturally structured with intact cell-to-cell contacts in control samples. Smooth and regular cells with intact cell membranes were observed in both the control (Figure 5A,D) and the quercetin-supplemented groups (Figure 5B,C,E,F). The rough and uneven appearance of quercetin-treated bacterial cells indicated that the cells had lost their usual shape (Figure 5C,F).

### 3.4. Motility, Biofilm Forming, Virulence, and QS Sensing Relative Gene Expression Pattern

Figure 6 shows the expression of *V. parahaemolyticus* flagella motility (*flaA* and *flgL*), biofilm formation (*vp0952* and *vp0962*), and QS (*luxS* and *aphA*) determined by RT-PCR in the sub-MIC of quercetin (from 0 to 110 µg/mL). At the various sub-MIC concentrations of quercetin, gene expressio was considerably downregulated (*p* < 0.05). This section may be divided by subheadings. It should provide a concise and precise description of the experimental results, their interpretation, as well as the experimental conclusions that can be drawn.

## 4. Discussion

Natural chemicals originating from plants may provide a potentially feasible strategy to bypass bacterial biofilm inhibitory processes and restore quercetin efficacy. Plant extracts, which contain quercetin, could be regarded as food ingredients rather than food additives. Quercetin is a non-specific protein kinase enzyme inhibitor. The FDA approved the use of high-purity quercetin at quantities of up to 500 milligrams (mg) as an ingredient in a variety of food categories in 2010 [33]. The goal of the current investigation was to determine whether quercetin at sub-MIC levels could be used to inhibit the growth of *V. parahaemolyticus*. Against *V. parahaemolyticus*, quercetin has antibacterial efficacy, which we describe in our study. We revealed that there was a dose-dependent bactericidal effect of quercetin against *V. parahaemolyticus* as well as a considerable biofilm formation inhibition caused by quercetin using a variety of techniques, including bacterial motility and growth of biofilm. Quercetin not only inhibited bacterial growth but also suppressed pathogenicity, biofilm formation, flagella motility, and QS gene expression in response to *V. parahaemolyticus*. 

The MIC of quercetin against *V. parahaemolyticus* ATCC27969 was determined to be 220 g/mL. The quantity of quercetin varies according to species. Quercetin is a pentahydroxyflavone, with the five hydroxy groups placed at the 3-, 3′-, 4′-, 5- and 7-positions. It is one of the most abundant flavonoids in edible vegetables, fruit, and wine. It has a role as an antibacterial agent, an antioxidant, a protein kinase inhibitor, and an antineoplastic agent. High concentrations of quercetin consist of more active compounds to increase the activity and bind with a specific portion to increase scavenging of ROS [48]. The MIC was established to be the lowest quantity that completely inhibited visible growth. The MIC of quercetin was determined as 80 µg/mL for *P. aeruginosa* and *K. pneumoniae*, 120 µg/mL for *Chromobacterium violaceum*, 250 µg/mL for *S.* Typhimurium, and 95 µg/mL for *Yersinia enterocolitica* [33,41,65]. By encouraging surface adhesion, swimming and swarming locomotion affect bacterial biofilm development. Our results clearly show that quercetin dramatically decreased the test pathogens’ flagella-mediated motility when compared to the control (Figure 1 and Figure 2). The outcomes are analogous to those reported by Damte et al. [66], who found that plant extracts can reduce *Pseudomonas* swarming motility by 71%. Another finding was that cinnamaldehyde prevented *E. coli* swarming by reducing biofilm development, according to Niu and Gilbert [67]. Similarly, quercetin reduced the motility at swimming (77 and 76%) and swarming (55 and 54.5%) against *S. typhimurium* [33,41]. As a result, quercetin seems to inhibit the ability of food-borne pathogens to attach to surfaces; hence, reducing the formation of biofilms. Additionally, bacterial motility, including swimming and swarming, is regarded as a key component of pathogenicity. In this case, quercetin significantly decreased the motility of the tested microorganisms. 

The development of biofilms is among the most important factors in a food-borne bacteria’s pathogenicity. QS is an important factor in the formation of biofilms [68]. Thus, disrupting the signal-mediated QS system may control the development of biofilms. The study’s findings demonstrated that quercetin effectively decreased the biofilm development in test pathogens at all tested concentrations. Our results are in line with those previously reported [33,41], which claimed that, as compared to control, quercetin (125 µg/mL)-treated food-borne pathogens *S. typhimurium* rarely form biofilms on food and food-contact surfaces. As previously reported [69], 0.051 mg/mL of quercetin used against *Listeria monocytogenes* biofilm formation and inhibited by quercetin [33]. In order to rule out any interference from quercetin (0.051 mg/mL) on planktonic populations during the experiment, its impact on *L. monocytogenes* planktonic growth kinetics was also assessed [33]. Because planktonic cells in the bulk medium continuously deposit onto layers of attached cells throughout normal development, it is important to recognize their role in biofilm formation. The results showed that the flavonoid quercetin prevented the development of *L. monocytogenes* biofilm and suggests that quercetin affects biofilm formation mechanisms other than cell division [33,69]. However, increasing quercetin levels had an impact on the formation of biofilms, as 1.96 and 3.21 Log10 CFU/cm^2^ of viable surface-associated cells were decreased at concentrations of 0.051 and 0.102 mg/mL, respectively, with a significant reduction (*p* < 0.05) in quercetin levels [33,69]. Additionally, at sub-MIC of quercetin, the biofilm was more inhibited by quercetin on food (shrimp and crab) (Figure 3 and Figure 4). *Vibrio* can attach to surfaces and form a biofilm, making the use of plastic cutting boards and cooking raw foods extremely prone to cross-contamination [33,70,71]. Additionally, compared to glass and SS surfaces, which are hydrophilic materials, plastic is more likely to allow *Salmonella* germs to stick to them [33,41,72]. Therefore, it is crucial to avoid contaminating the plastic cutting boards used while preparing or processing food because this leads to vibriosis. Other authors looked at the efficacy of quercetin to inhibit the formation of biofilms in *S. epidermidis* [45]. Quercetin inhibited the growth of biofilms in a concentration-dependent manner. Quercetin reduced the growth of *S. epidermidis* biofilm by 90.5 and 95.3% at 250 and 500 µg/mL concentrations, respectively [45]. For this reason, quercetin is a common source for the food and pharmaceutical industries [47]. Consequently, there is a growing market for biodegradable polymers (fibers) that are effective against microbes. *S. aureus*, *E. coli*, and *K. pneumoniae* were all susceptible to the antibacterial effects of polylactide-based polymers (fibers) loaded with quercetin-(Q)- [49]. In cellular and animal models of Alzheimer’s disease, nanoparticles of quercetin (nanoquercetin), which are colloidal in nature and made of polyaspartic acid-based polymer micelles encapsulated with quercetin, were utilized to prevent intracellular polyglutamine aggregation [50]. Quercetin that has been microencapsulated in a surfactant polymer capable of acting as an antisolvent process has been shown to be effective against cancer, develop biodegradable nanoparticles, and improve quercetin delivery [51,52].

We investigated biofilm generation and morphology by FE-SEM to further probe perturbations of the biofilm following quercetin application. The presence of quercetin decreased the number of connected cells and hampered the formation of biofilm, according to FE-SEM observation (Figure 5). As previously reported, quercetin has effects on *S. typhimurium* and *V. parahaemolyticus* biofilms as confirmed by FE-SEM [7,33,41]. In our study, 1/2 MIC showed inhibition was increased compared to the control and 1/4 MIC (Figure 5). 

Many genes are essential for the pathogenicity, biofilm development, and physiological traits of *V. parahaemolyticus*. We observed the *V. parahaemolyticus* gene expression profiles for QS (*luxS* and *aphA*), motility (*flaA* and *flgL*), and virulence (*vp0952* and *vp0962*) in order to evaluate the potency of quercetin. There are connections between pathogenicity, QS, and virulence elements procedures. An emerging technique for preventing biofilm development, minimizing pathogenic infections, and maintaining food safety is to prevent or limit QS production. Oxidative stress develops when ROS build up occurs inside the cell [73]. By enhancing microbial population adaptation and survival protection, oxidative stress contributes significantly to the production of biofilms [53]. Not just in human cells, but also in microbes, ROS are crucial signaling molecules [41]. To keep a healthy redox cycle going and to encourage microbial adhesion, ROS can act as both intracellular and extracellular stimulants [33,46]. This will eventually result in the formation of biofilms. There may be an accumulation as a result of a disruption in the redox cycle [33]. By scavenging ROS within cells and weakening the membrane integrity of bacterial cells, the antioxidant quercetin prevents the formation of biofilms [46]. Quercetin, which acts as a potent bioactive compound, has antioxidant properties that lead to scavenging of the ROS production. Quercetin significantly reduced both forms of motility as well as the transcription of the *flaA* and *flgL* genes in the current investigation. These genes are connected to the control of flagella synthesis and structure in *V. parahaemolyticus* [7]. For instance, the *flaA* gene, which encodes polar flagellin, contributes to swimming motility, and the lateral flagellar gene system of *V. parahaemolyticus*, and allows bacteria to spread out and colonize surfaces (swarming), contains the *flgM* gene, which encodes anti-28 [7,74]. These results were in line with those of an earlier study [75], which found that thymoquinone decreased the expression of genes related to flagella production and hindered the motility of *V. parahaemolyticus*. A number of virulence factors, in addition to adhesion, are involved in the pathogenesis of *V. parahaemolyticus*, and their expressions affect the pathogen’s pathogenicity. Specifically, for the *vp0952* and *vp0962* genes, our findings showed that quercetin significantly reduced the expression of a number of virulence genes (Figure 6). On chromosome 2 of *V. parahaemolyticus*, the genes *vp0950*, *vp0952*, and *vp0962* all encode proteins that are similar to those found in biofilms [7,76]. Natural plant extracts also dramatically downregulated the transcription of the genes *ompW*, *luxS*, and *aphA*, which had previously been downregulated by citral in a prior study [7,77]. The quorum-sensing regulation, a challenging cell-to-cell process that enables bacteria to monitor their surroundings and cooperate, is primarily regulated by the two genes of *luxS* and *aphA*, which have been extensively investigated [7,78]. The regulation of thermostable direct hemolysin (TDH) and the development of *V. parahaemolyticus* biofilms have also been reported to be regulated by the *luxS* gene [7,79]. The impact of quercetin on other virulence genes, however, has to be further investigated as the current research largely focused on the alterations in biofilm-related genes.

Quercetin is not likely to enter cells and directly interact with transcriptional regulators or intracellular objectives. Our hypothesis is that quercetin might interact with specific membrane proteins, activating the bacterial signaling system and resulting in transcriptional changes that result in the downregulation of genes. In addition to microbial adhesins and cell membrane proteins, quercetin is a polyhydroxy hydrolytic chemical that has the potential to form powerful complexes with a wide range of macromolecules. By adjusting to the changes in the membrane, the bacterial cells may alter how their genes are produced using bacterial signaling processes, such two-component systems.

## 5. Conclusions

As a result, we were able to demonstrate that quercetin has effective antimicrobial and maybe anti-pathogenicity properties against *V. parahaemolyticus* on shrimp and crab shell surfaces. Furthermore, quercetin (1/2 MIC) significantly reduced the number of viable bacterial cells on shrimp and crab shell surfaces (3.70 and 3.09 log CFU/cm^2^, respectively), disrupted cell-to-cell connections (FE-SEM images), dislodged already-formed biofilms, and significantly reduced the expression of genes associated with motility, virulence, and QS (Figure 6). Quercetin may be developed as an alternate technique to manage the biofilm of *V. parahaemolyticus* in food systems and lower the risk of foodborne illness caused by this pathogen.

## Figures and Tables

**Figure 1 polymers-14-03847-f001:**
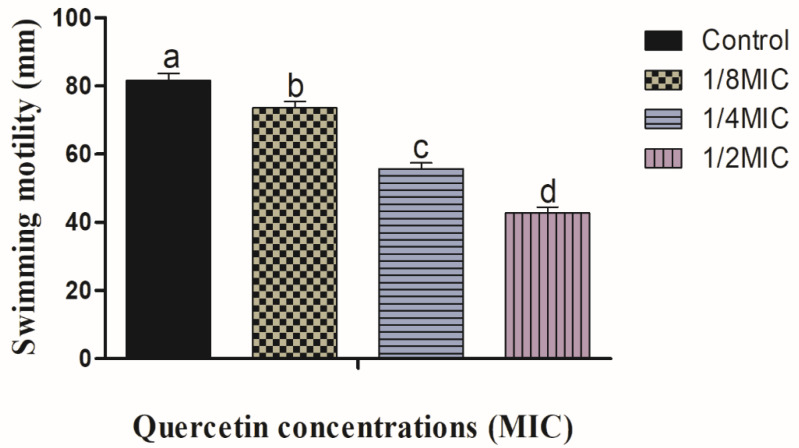
Swimming motility assay for *Vibrio parahaemolyticus* with sub-MICs of quercetin (μg/mL). Data represented as mean ± SEM of three independent replicates. ^a–d^ Values with different letters differ significantly by Duncan’s multiple-range test (*p* < 0.05).

**Figure 2 polymers-14-03847-f002:**
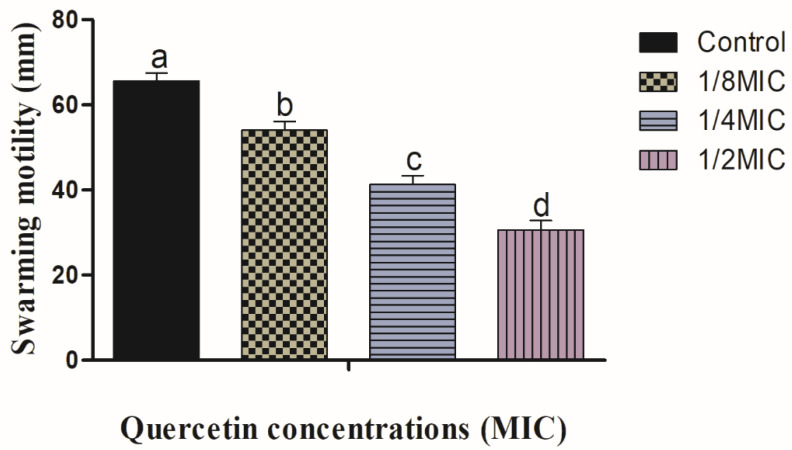
Swarming motility assay for *Vibrio parahaemolyticus* with sub-MICs of quercetin (μg/mL). Data represented as mean ± SEM of three independent replicates. ^a–d^ Values with different letters differ significantly by Duncan’s multiple-range test (*p* < 0.05).

**Figure 3 polymers-14-03847-f003:**
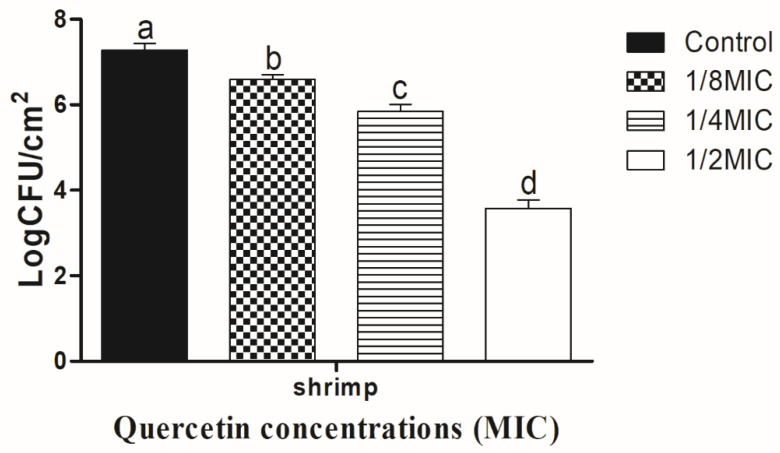
Inhibition of *Vibrio parahaemolyticus* biofilm formation (24 h) on shrimp surfaces by sub-MICs of quercetin (μg/mL). Data represented as mean ± SEM of three independent replicates. ^a–d^ Values with different letters differ significantly by Duncan’s multiple-range test (*p* < 0.05).

**Figure 4 polymers-14-03847-f004:**
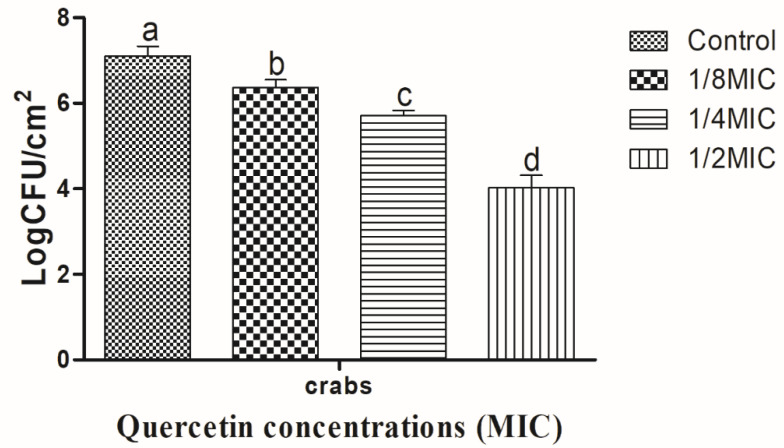
Inhibition of *Vibrio parahaemolyticus* biofilm formation (24 h) on crab surfaces by sub-MICs of quercetin (μg/mL). Data represented as mean ± SEM of three independent replicates. ^a–d^ Values with different letters differ significantly by Duncan’s multiple-range test (*p* < 0.05).

**Figure 5 polymers-14-03847-f005:**
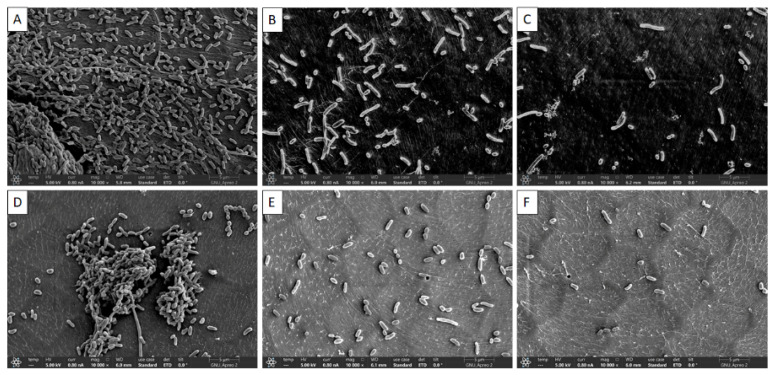
Representative scanning electron micrographs of *Vibrio parahaemolyticus* biofilms formation in the presence of sub-MICs of quercetin on the crab surfaces: (**A**) Control (0% quercetin); (**B**) 1/4 MIC; (**C**) 1/2 MIC and shrimp surfaces; (**D**) Control (0% quercetin); (**E**) 1/4 MIC; (**F**) 1/2 MIC.

**Figure 6 polymers-14-03847-f006:**
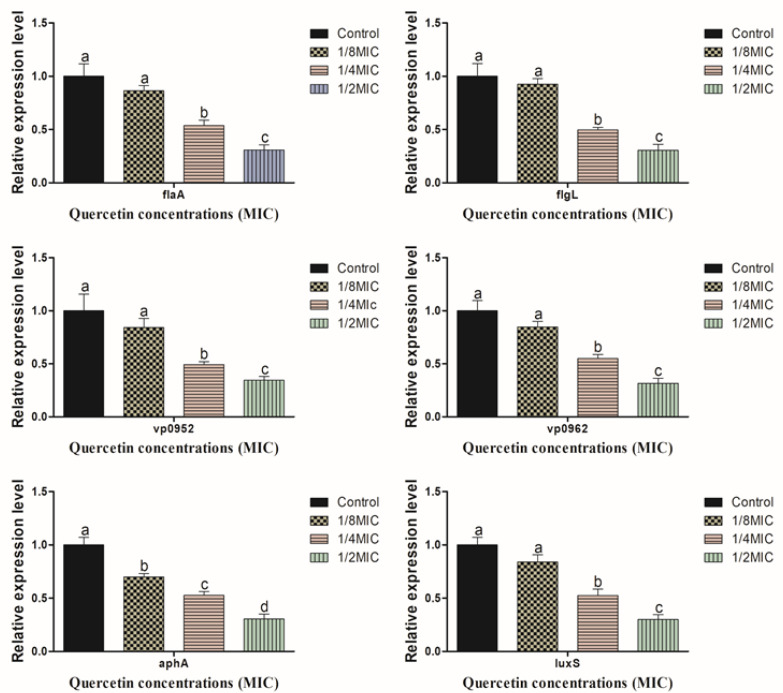
Relative expression levels of *flaA, flgL, vp0952, vp0962, aphA*, and *luxS* genes in *Vibrio parahaemolyticus* supplemented with sub-MICs of quercetin. ^a–d^ Different superscript letters indicate significant differences (*p* < 0.05) with three independent replicates.

**Table 1 polymers-14-03847-t001:** Primer lists used in this study for RT-qPCR. F and R stand for forward and reverse primers.

Target Gene	Sequence of Primers (5′-3′)	Product Size (bp)	NCBI Accessions No.
*flaA*	F: CGGACTAAACCGTATCGCTGAAA R: GGCTGCCCATAGAAAGCATTACA	128	GQ433373.1
*flgL*	F: CGTCAGCGTCCACCACTT R: GCGGCTCTGACTTACTGCTA	141	CP066246.1
*luxS*	F: GGATTTTGTTCTGGCTTTCCACTT R: GGGATGTCGCACTGGTTTTTAC	119	CP066246.1
*aphA*	F: ACACCCAACCGTTCGTGATG R: GTTGAAGGCGTTGCGTAGTAAG	162	CP066246.1
*vp0952*	F: TATGATGGTGTTTGGTGC R: TGTTTTTCTGAGCGTTTC	276	CP064041.1
*vp0962*	F: GACCAAGACCCAGTGAGA R: GGTAAAGCCAGCAAAGTT	358	CP064041.1
*16S rRNA*	F: TATCCTTGTTTGCCAGCGAG R: CTACGACGCACTTTTTGGGA	186	CP085308.1

## Data Availability

The data presented in this study are available on request from the corresponding author.

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
