# Peer review of "Antimicrobial Efficacy of Quercetin against Vibrio parahaemolyticus Biofilm on Food Surfaces and Downregulation of Virulence Genes"

_polymers, 2022, doi:10.3390/polym14183847_

Round 1
Reviewer 1 Report
This interesting work presents the results of quercetin antibacterial action, with vey good analysis of the biofilm formation and colony formation prevention.
Although this work can be very interesting for microbiology specialists and reseaechers who are working in the field of antibacterial coatings and surfaces, this paper seemed to be not suitable for the publication in the "Polymers" journal.
Indeed, this work could be more appropriate for biology/microbiology specialized journals or at least applied chemistry journals. In the whole manuscript the term POLYMER appeared only once, when the definition of the biofilm was given.
I suggest to resubmit this paper to the more appropriate journal.
Minor suggestion:
1) The effect for the 1/2 MIC dosage was lower than one could expect, can you show the results for 1 MIC dosage?
2) Can you provide surface characterization of the quercetin-modified surfaces?
Author Response
Thank you respected reviewer for your helpful information. Please see the attachment.

Reviewer 2 Report
It is well-marked that this paper would have a little formally errors.
Please convert to “P” from “p” the statistical significant symbol.
Also, I would like to some humble suggestions to improve your the manuscript well.
Please see some papers to improve your article well and you can use them in the introduction section. It can be improved some sentences related to fish and sea foods topics.
Please add these sentences in the first paragraph of the introduction section.
"Nutritious and tasty aquatic products are susceptible to oxidation and bacterial contamination in planned transportation. This situation damages the taste of aquatic products and seriously threatens food safety. Since seafood is freshly used, the food quality is very high, but by time its quality will decline and become worsen for consumption. The quality and safety of seafood is a major challenge facing its own seafood related industry in food sciences, and mostly in fisheries and aquaculture research departments. Fish is an essential nutrient in the human diet and is also present in the global aquatic product industry for consumers. Such products are prone to oxidation and bacterial contamination in organized transportation, which not only destroys the taste of products, but also poses a very serious threat to food safety" (Selamoglu, 2021).
-Mesut Selamoglu. Importance of the cold chain logistics in the marketing process of aquatic products: An update study. Journal of Survey in Fisheries Sciences 8(1) 25-29 2021.
It can be improved some sentences related to the properties of the quercetin .
Therefore, Please see the paper to cite below in the introduction section:
-OZGEN SENAY, KILINC OK, SELAMOGLU ZELIHA. Antioxidant Activity of Quercetin: A Mechanistic Review. Turkish Journal of Agriculture - Food Science and Technology. 4(12): 1134-1138, 2016.
It is well-marked that this paper would may be an incremental contribution of the manuscript to the field. It is very well-marked that this paper is acceptable with minor revision and useful for publish in this journal. |
Author Response
Thank you dear respected reviewer for your useful information. Please see the attachment.

Reviewer 3 Report
This article contains adequate data but some errors need to corrected, such as bacterial names are not italic. Additionally, authors did not describe clearly how to treat biofilm cells with quercetin.
Line 28: V. parahaemolyticus is not italic.
Line 38-39: According to studies by Han et al. [3] and Almo-hamad et al. [4] over 60% outbreaks by V. parahaemolyticus biofilm. –it is not a correct English.
Line 58-59: A reference is better to back up this statement.
Line 120: 105 log CFU?
Line 122 105 CFU?
Line 140: 105 log CFU/mL?
Line 145: why used CHROMagar that is a selective medium and could inhibit the growth of partially injured bacteria unless authors used non-sterile 96-well plates.
Line 155: Quercetin was added before or after autoclave?
Line 161: control, was the control no quercetin or at full strength of MIC? Need to be clarified.
Section 2.5: authors did not describe that procedures of treating biofilm by quercetin? “Control, 1/8, 1/4, and 1/2 MIC concentrations were used in this study. In a 50 mL conical 161 tube with 10 mL TSB, quercetin, and 100 μL of bacterial suspension (105 log CFU/mL), the 162 prepared samples were placed.” There was no biofilm involved. “Each coupon 166 (shrimp and crab) was placed in a sterile stomacher bag with 10 mL of peptone water (PW; 167 BD Diagnostics, Franklin Lakes, NJ, USA), and processed using a Stomacher (Bag Mixer; 168 Interscience, Saint Nom, France) at the highest speed of 4 for two min to release the bac-169 teria that form biofilms on the coupons [33].” There was no quercetin involved.
Line 180-181: suggesting revised this sentence.
Line 186-189: The cells added into the Falcon tubes were not biofilm cells. Why did not authors use the biofilm on coupons for this test?
Line 289, 297, 309: Salmonella Typhimurium?? Typhimurium should not be italic.
Line 310, 312, 320, 321: suggesting convert 0.2 mM into mg since the unit used in this study was mg.
Line 326: Salmonella
Line 329: Staphylococcus
Line 351: By generating ROS within cells and weakening the membrane integrity of bacterial cells, the antioxidant quercetin prevents the formation of biofilms [45]. It is contradictory description. Authors also describe in “Introduction”, quercetin is an antioxidant. Thus, it is able to reduce ROS. However, here says “generating ROS”.
Author Response
Thank you respected reviewer for your useful information. Please see the attachment.

Reviewer 4 Report
An interesting knowledge has been proposed. However the following comments should be addressed before acceptance
Novelty of the manuscript must be better emphasized
Significance of this paper should be clearly mentioned in Introduction part
How does Concentration effects the biological activity
1. Rephrase the conclusion with highlighting results, conclusions should be ameliorated by including more numerical data.
update references
Author Response

(The authors gave the same response as above.)

Round 2
Reviewer 1 Report
The authors have answered to my questions, especially to the point 1, regarding the correlation with the Polymers audience. The answers are reasonable and I suggest to accept this paper for this Special Issue.
Reviewer 3 Report
all the points have been correctly revised.